https://doi.org/10.1038/s41467-020-16494-0　　**OPEN**

# Continuous crystalline graphene papers with gigapascal strength by intercalation modulated plasticization

Peng Li[1], Mincheng Yang[1], Yingjun Liu[1], Huasong Qin[2], Jingran Liu[2], Zhen Xu [1]✉, Yilun Liu[2]✉, Fanxu Meng[1], Jiahao Lin[1], Fang Wang[1] & Chao Gao[1]✉

Graphene has an extremely high in-plane strength yet considerable out-of-plane softness. High crystalline order of graphene assemblies is desired to utilize their in-plane properties, however, challenged by the easy formation of chaotic wrinkles for the intrinsic softness. Here, we find an intercalation modulated plasticization phenomenon, present a continuous plasticization stretching method to regulate spontaneous wrinkles of graphene sheets into crystalline orders, and fabricate continuous graphene papers with a high Hermans' order of 0.93. The crystalline graphene paper exhibits superior mechanical (tensile strength of 1.1 GPa, stiffness of 62.8 GPa) and conductive properties (electrical conductivity of $1.1 \times 10^5$ S m$^{-1}$, thermal conductivity of 109.11 W m$^{-1}$ K$^{-1}$). We extend the ultrastrong graphene papers to the realistic laminated composites and achieve high strength combining with attractive conductive and electromagnetic shielding performance. The intercalation modulated plasticity is revealed as a vital state of graphene assemblies, contributing to their industrial processing as metals and plastics.

[1] MOE Key Laboratory of Macromolecular Synthesis and Functionalization, Department of Polymer Science and Engineering, Key Laboratory of Adsorption and Separation Materials & Technologies of Zhejiang Province, Zhejiang University, 38 Zheda Road, 310027 Hangzhou, P. R. China. [2] State Key Laboratory for Strength and Vibration of Mechanical Structures, School of Aerospace, Xi'an Jiaotong University, 710049 Xi'an, P. R. China. ✉email: zhenxu@zju.edu.cn; yilunliu@mail.xjtu.edu.cn; chaogao@zju.edu.cn

Graphene with two-dimensional (2D) topology has two distinct aspects of properties: the unprecedented mechanical and conductive properties in the planar direction yet a considerable softness with low out-of-plane bending rigidity[1–3]. On the one side, the planar carbon bonding of graphene renders extremely high mechanical strength (130 GPa), record thermal conductivity (5300 W m$^{-1}$ K$^{-1}$) and superior electrical conductivity (10$^8$ S m$^{-1}$), all along its basal plane[4]. These superior in-plane properties have been expected to be translated into outstanding performances of their macroscopic materials. The ideal philosophy is fashioning individual graphene sheets into a crystalline order through fluid processing, achieving the utmost efficiency to express their favorable properties in macroscopic materials. For examples, graphene papers (GPs) with a lamellar structure can be made by infiltration-aided assembly[5–11], solution casting[12–17] and spray coating[18,19]; graphene fibers with regular alignment have been fabricated by wet-spinning of graphene oxide (GO) liquid crystals[20]. On the other side, the considerable softness—the other hidden attribute of graphene—is prone to degrade the crystalline order of graphene assemblies, because of the easy formation of chaotic wrinkles in the assembly process from dispersions to solid materials[16]. Single-layered graphene has a bending rigidity of 40−80 kT, which is softened by defects, down to 1 kT for GO sheet[3,21]. During the fluid assembly, the high softness of GO sheets aggravates the tendency to form random wrinkles in solid materials, driven by the slow dynamic motion, skinning and capillary contraction in drying[22]. This is the very reason that random wrinkles are ubiquitous in macroscopic graphene materials, which result in their unfulfilled performances, for instance, low mechanical strength around 453 MPa of pure GO papers (GOPs) and 660 MPa of pure chemically reduced GPs[17,18].

Wrinkles can enhance the flexibility and porosity of graphene materials, which are useful for flexible electronic devices and energy storage[16,23]. However, the existence of wrinkles inevitably depresses the transportation of electrons and phonons of individual graphene sheet, and hinders the compact and regular alignment of graphene assemblies. In the wet transfer of single-layer graphene films, selected substrates to match the crystal lattice flattens ripples and wrinkles of graphene, achieving extremely high electron transport properties[24]. For the macro-assembly, random wrinkles spontaneously form in fluid processing of massive graphene sheets, bringing severe stress concentration and degrading the mechanical strength of the macroscopic assemblies, as well as electrical/thermal conductivities[22]. Previous efforts have focused on manipulating the ordering of fluid states to promote the alignment of graphene sheets. For instance, Dikin et al.[5] adopted a flow-aided infiltration method to make GOPs with a strength of 120 MPa. Zhong et al.[18] invented a continuous centrifugal spraying method to increase the alignment of GO laminates and improved the strength of GPs to 660 MPa. Careful analysis tells that massive wrinkles gradually generate by capillary force in the final evaporation of solvent. These defective wrinkles born in the solution process become a hidden barrier to achieve the ideal crystalline order of graphene assembly. However, how to eliminate the wrinkles of 2D graphene sheets in the solid laminates still remains a grand challenge.

Here, we propose an intercalation modulated plasticization (IMP) stretching method to regulate the intrinsic wrinkled conformation of 2D GO sheets into a high crystalline order in macroscopic papers. We reveal the brittle GO laminate turns to a plastic state with more than 230% enhancement of strain limit, resulting from the intercalation of solvent molecules. The plastic processing state allows the continuous stretching to flatten the random wrinkles in laminates, achieving a high Hermans' order parameter (*f*) of 0.93 and stacking density of 1.82 g cm$^{-3}$. The IMP stretching-induced crystallization is revealed at multiscales by online synchrotron X-ray scattering method and high-resolution transmission electron microscope (TEM). The promotion of crystalline order reduces stress concentration sites and electron/phonon scattering centers, and thus endows GPs with record performances, including the superb strength of 1.1 GPa, high stiffness of 62.8 GPa, outstanding thermal/electrical conductivities of 109.11 W m$^{-1}$ K$^{-1}$ and 1.09 × 10$^5$ S m$^{-1}$, respectively. High-performance GPs are continuously produced with meter-scale length. Profiting from the high output, we produce lamellar composites with high enough structural strength of 634 MPa and attractive electrical conductivity of 3.1 × 10$^4$ S m$^{-1}$, extending the use of graphene for realistic uses.

## Results

**The continuous fabrication of crystalline graphene papers**. We set up a continuous plastic stretching crystallization process to turn the continuous direct-cast GOPs with chaotic wrinkles into highly crystalline GOPs, as shown in Fig. 1. This process includes three vital steps: (1) IMP to trigger the plastic transition. Direct-cast GOPs were immersed into the plasticization bath (e.g. ethanol) and transited from rigid to soft state as the solvent molecules were intercalated into the interlayer space. (2) Plasticization stretching to eliminate chaotic wrinkles. Tuning the speed ratio ($V_{\text{T}}/V_0$) between the two rollers can afford variable stretching ratio (SR = $V_{\text{T}}/V_0$) from 0 to 8%. (3) Drying under a tension, immobilizing the extended conformation in solid GOPs. Followed by chemical reduction (see "Methods"), GPs were obtained and labeled as SR-GPs, for instance, 8%-GP means GP with an SR of 8% in the IMP stretching process. Different from other previous methods to improve the alignment of GOPs and GPs, such as infiltration-aided assembly and centrifugation sparying[5,18], our method gives an alternative poststrategy to engineer the wrinkles spontaneously formed in the final drying, which has been ignored previously, and completes the structural control of GPs starting from the dispersed state to the final solid state. Importantly, our simple process is convenient to continuously fabricate crystalline GPs up to meters length (Fig. 1b), which can be directly combined into an industrial production line.

**Intercalation modulated plasticization mechanism**. Direct-cast GOPs show a nearly elastic deformation and negligible plasticity with a tensile strain of only 3%. This intrinsic brittleness of GO bulks makes it impossible to eliminate random wrinkles into a crystalline order. We observed that direct-cast GOPs have spreading wrinkles on the surface and sections, and these wrinkles remain after elastic breakage (Supplementary Fig. 6b−e), denying the engineering of wrinkles in the elastic state. To eliminate these wrinkles, we reason that attaining plastic state is imperative to mobilize GO sheets for re-arrangement, inspired by the traditional polymer processing to trigger the plastic state by heating or solvent swelling.

We found an obvious brittle-plastic transition of GOPs as immersed in polar solvents. The ultimate tensile strain of GOPs, an intuitive parameter to evaluate the plastic deformation, reaches more than 10% in ethanol, 230% higher than that (3%) of direct-cast GOPs. Accompanying with the plastic transition, the interlayer space of GOPs expands. Ethanol molecules are intercalated into GO laminates to attain a saturated state, with the interlayer distance increasing from 0.90 to 1.58 nm, calculated from (001) peak in X-ray diffraction spectra (XRD; Fig. 2a). According to Lifshiz's formula, the expanding interlayer weakens the van der Waals interaction by intercalating plasticizers and

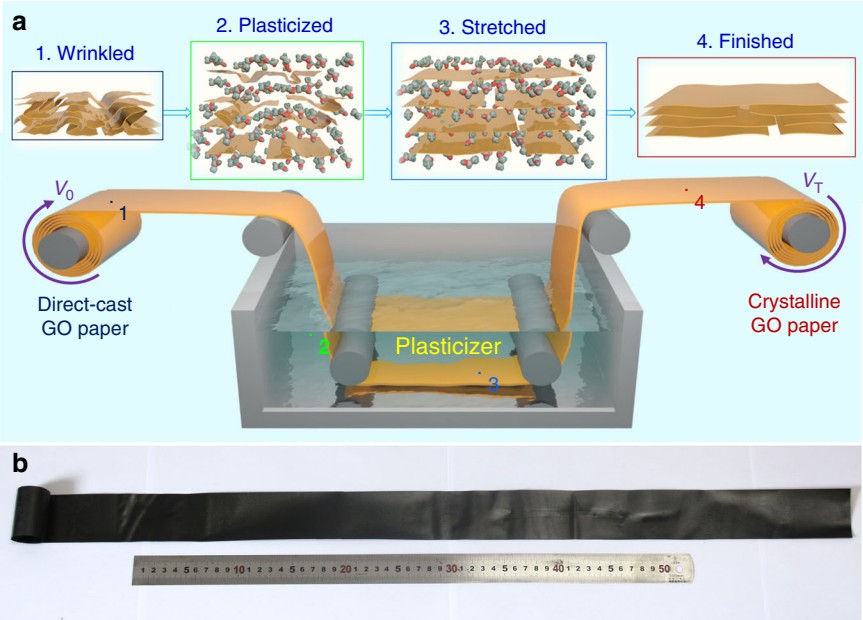

**Fig. 1 Production of highly crystalline graphene oxide papers by plasticization stretching method. a** Schematic of the continuous IMP stretching apparatus. **b** Photograph of the meter-long GOP after continuous IMP stretching.

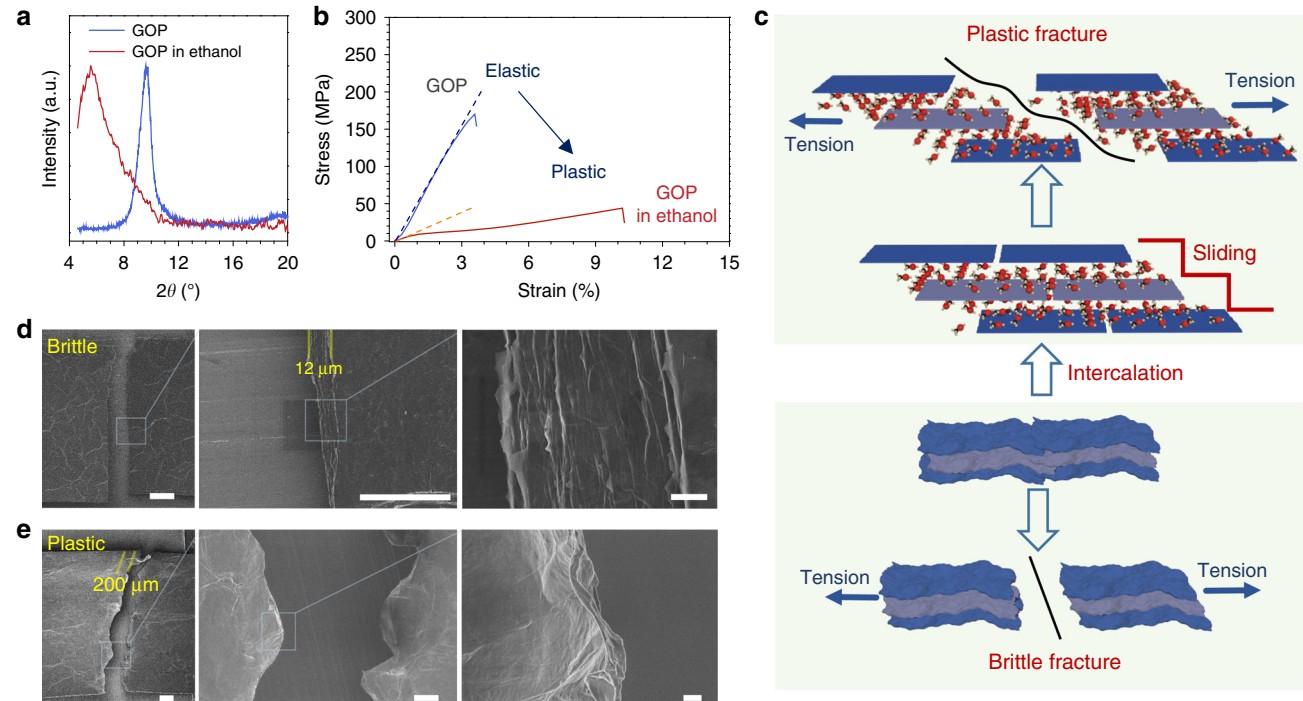

**Fig. 2 Intercalation modulated plasticization of graphene oxide papers. a** XRD profiles of the direct-cast GOP and the plasticized GOP with ethanol intercalating. **b** Typical tensile curves of the plasticized GOP and the direct-cast GOP, showing a distinct transition from elastic to plastic deformation. **c** Schematic of the plastic sliding of GO sheets induced by the increased interlayer distance and typical plastic and brittle fractures. **d, e** The fracture of direct-cast GOPs (**d**) and plasticized GOPs (**e**), respectively. Scale bar in (**d, e**), 200 μm (left), 50 μm (middle), and 2 μm (right).

activates the sheet sliding under tension (Fig. 2c; Supplementary Fig. 2)[25], in analogy with the thermal plastic mechanism of linear polymers[26]. We rationalized the elastic−plastic transition by a nonlinear spring-bead model with the concept of self-healable interlayer interactions (e.g., hydrogen bonding), based on a previously reported deformable tension-shear (DTS) chain model (Supplementary Fig. 3)[27,28]. From this model, their failure and reconstruction of interlayer links initiating at the end and then

propagating to the middle of laminated GO sheets afford the plastic deformation. As the interlayer spacing expands after solvent intercalation, the failure and reconstruction of the weakening interlayer links continuously occur in balance, contributing to the plastic strain of GOP as the dislocation gliding in metals. As a result, a large amount of energy is dissipated and the plastic limit strain is significantly increased as observed in the tensile tests of GOPs (Fig. 2b). We numerically

simulated the tensile behavior of GOPs with ethanol intercalated in 1.58 nm interlayer spacing. The simulated tensile curve exhibits a typical plastic deformation trend with a high breakage elongation of 10%, which coincides well with the experimental result (Supplementary Fig. 3b).

The activated sheet sliding by solvent intercalation was verified in the fracture morphology analysis by scanning electron microscope (SEM)[29,30]. The fracture of a plasticized GOP exhibits an "H" shape and features three important characterizations of plastic fracture, which are fibril-like plastic orientation, near 45° shearing deformation, and fracture retraction (Supplementary Fig. 4). By contrast, dried GOPs exhibit a straight crack across the paper, denoting a brittle fracture. For plasticized GOPs, the fracture width reaches 200 µm. But for direct-cast-dried GOPs, the fracture width is only 12 µm (Fig. 2d, e). This comparison demonstrates that the interlayer slippage is activated by solvent intercalation, which determines the plastic transition of GOPs.

**Plasticization stretching-crystalline process.** The plastic transition of GOPs allows the activated sheet sliding and thus the flattening of wrinkles by stretching. Real-time polarizing optical microscope (POM) and small angle X-ray scattering (SAXS) measurements demonstrate that random wrinkles in direct-cast GOPs are stretched to become an extended state to reach a high crystalline order (Fig. 3a). POM tracking directly shows the diminishment process of wrinkles with the increased tensile strain ($\varepsilon$) from 0 to 7%. Figure 3b shows the wrinkles perpendicular to the stretching direction (labeled as Type II wrinkles) first weaken at $\varepsilon = 2$ and 5%, and finally disappear at $\varepsilon = 7\%$. With $\varepsilon$ increasing, the equatorial streak scatterings become stronger gradually and $f$ increases from 0.82 to 0.87 (Fig. 3c). This increasing $f$, calculated from the azimuthal scan of SAXS patterns (see "Methods"), indicates that the orientation of structural units of GOPs at nanoscale is gradually improved as tensile strain increases[31].

**Crystalline order of graphene papers.** The fully extended structure of GO sheets by continuous stretching crystallization is maintained in the GPs after chemical reduction by hydroiodic acid (HI). From XRD spectra, GPs have higher average crystalline degree as SR increases[32]. The interlayer distance decreases from 0.37 nm at 0% SR to 0.36 nm at 8% SR (Fig. 4a). The full width at half maximum (FWHM) of the (002) crystalline plane stepwise decreases from 3.35° of direct-cast GPs without stretching to 2.03° of 8%-GPs, corresponding to a 65% higher crystal thickness (3.96 nm) than that of direct-cast GPs (2.40 nm; Fig. 4b).

The improved crystalline order by IMP stretching reflects not only the stepwise increased crystalline degree, but also the orientation degree of structural units at multiple scales. Direct-cast GPs are full of random wrinkled textures on surface (SEM images at 0% SR in Fig. 4d), which inevitably cause stress concentration to degrade the mechanical strength[33,34]. As SR gradually increases to 2%, surface wrinkled textures fade to become smooth. At SR exceeding 5%, wrinkles parallel to the tension direction (type I) emerge on surface and wrinkles perpendicular to tension direction (type II) gradually disappear (Supplementary Fig. 6a), because of the flattening effect under tension. These emerged type I wrinkles and diminished type II wrinkles show that beyond the activated sheet sliding in molecular scale during deformation process, the wrinkles network on the surface deforms, too. We used SAXS and wide angle X-ray scattering (WAXS) measurements on the section parallel to the stretching direction to examine the packing order at micro and atomic scales, respectively[31,35] (Fig. 4c). GPs were revealed to have a hierarchical structure: graphene sheets stack to form a basic plate with nanometers thickness and these plates pile up to become papers with micrometers thickness (Supplementary Fig. 9)[36]. The anisotropic pores between basic plates can be detected by SAXS and their alignments equally reflect the ordering of plates. As shown in SAXS results in Fig. 4d, the $f$ of micropores in GPs was stepwise improved from 0.81 of direct-cast samples to 0.88 at an SR of 8%. For the packing order of graphene sheets inside the plates, WAXS tracking demonstrates that their order parameter was enhanced from 0.85 of direct-cast paper to 0.93 at an SR of 8%.

Different from the enhanced crystallinity in both direction of one-dimensional polymers and CNTs by stretching[37,38], the 2D topology of graphene sheets renders an unsymmetrical enhancement in packing order. Along the stretching direction, type II wrinkles are flattened and crystalline order is continuously enhanced. In the other perpendicular direction, $f$ increases slightly from 0.85 to 0.87 at an SR of 2% and keeps steady within 2−8% of SR. The beginning increase is caused by the

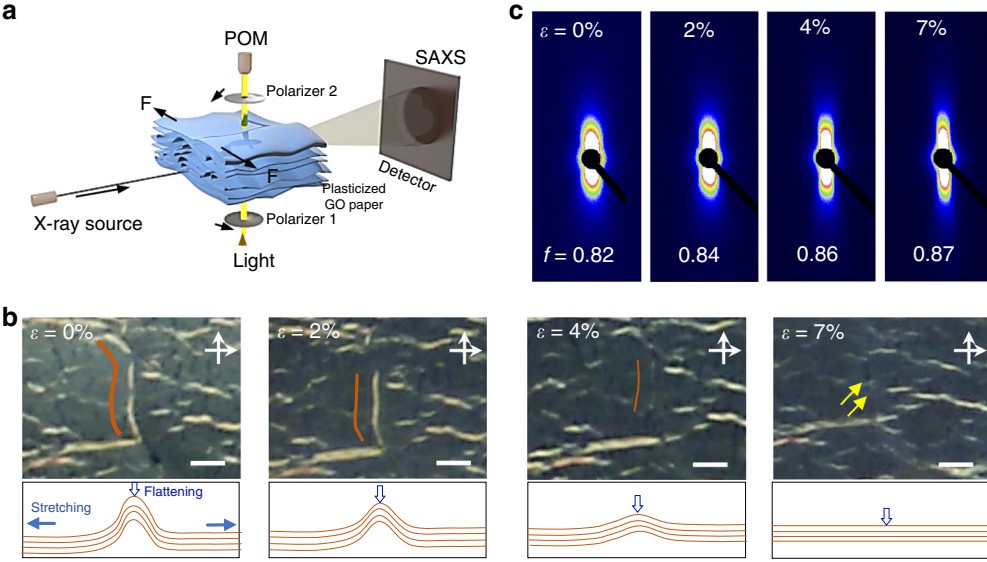

**Fig. 3 Crystalline evolution during plasticization stretching of plasticized graphene oxide papers. a** Schematic of real-time tracking of the structural change of the plasticized GOP during tension ($\varepsilon$). **b, c** POM images (**b**) and SAXS patterns (**c**) of the plasticized GOP at different $\varepsilon$. Scale bar, 50 µm (**b**).

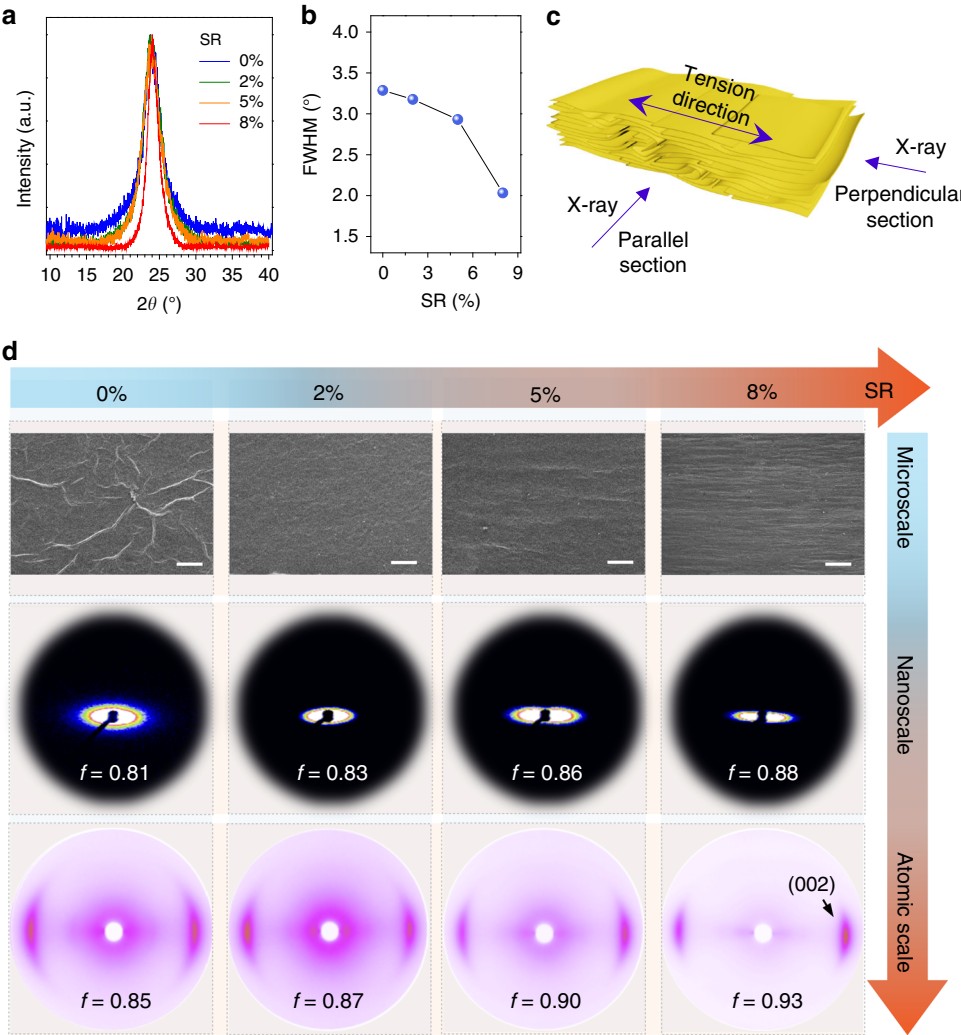

**Fig. 4 Crystalline order of graphene papers. a** XRD profiles of GPs at a serials of *SR*. **b** *FWHM* of (002) peak in (**a**), showing gradually aligned and homogenized crystalline structure with *SR* increasing. **c** Schematic of the SAXS and WAXS tests, in which parallel section represents that X-ray is exposed on the section parallel to the stretching direction, perpendicular section represents that X-ray is exposed on the section perpendicular to the stretching direction. **d** Oriented structures of GPs in micro-scale (upper row, SEM images), nanoscale (middle row, SAXS patterns) and atomic scale (bottom row, WAXS patterns). Scale bar, 100 μm (**d**).

negative Poisson ratio of wrinkles, which was observed in GOP recently[36]. After 2% SR, the Poisson ratio becomes normal positive and type II wrinkles become compressed in the perpendicular direction of stretching (Supplementary Fig. 4a). We use the ratio between parallel order and perpendicular order ($f_{\parallel}/f_{\perp}$) to evaluate the anisotropic enhancement and its value at an SR of 2% is identical with the unstretched sample and begins to increase from 1 at an SR of 2% to 1.07 at 8% (Supplementary Fig. 8a, b).

We used TEM to visually convince the stretching crystallization effect and verify the high crystalline order of stretched GPs at the atomic scale[39]. The sample was prepared by ion milling the GP embedded in epoxy resin to get intact slices on copper grids. Figure 5a, b shows that 8%-GPs feature an extremely extended state of graphene sheets and high crystallinity with $f$ of 0.95 calculated by fast Fourier transform (FFT)[40]. As a distinctive contrast, direct-cast GPs exhibit wave-like arrangement of graphene sheets and its $f$ is 0.90, which is in accordance with the WAXS results (Fig. 4d). Closer inspections reveal that the stacking disorders in direct-cast GPs, including crumples, folds, and dislocations, were rearranged to be in regular alignment by IMP stretching (Fig. 5b, d).

**Properties of graphene papers**. The IMP stretching achieved highly crystalline GPs and greatly improved their mechanical properties. The Young's modulus and ultimate failure strength monotonously increase as SR increases (Fig. 6a, b). The modulus of GPs at an SR of 8% was tested as 60.27 GPa, 693% higher than that (7.60 GPa) of direct-cast GPs without stretching. This measured modulus, reaching that of aluminum alloys[41], makes GP a good candidate of structural material. The stiffness reflects the regularity of the structural units, and we reveal that the modulus has a positive correlation with the sheet order parameters in WAXS tests (Fig. 6c). This trend exhibits that the wrinkled conformation is rearranged to extended state as SR increases, thus promoting the sheet orientation and the modulus of GPs.

Increasing SR also improves the tensile strength of GPs. The tensile strength of 8%-GPs was tested as 1.1 GPa, 370% higher than that of direct-cast GPs (234.16 MPa; Fig. 6a), which becomes a new record strength of neat GPs as compared with previous reports (up to 660 MPa)[18]. Considering the low density of GPs (measured as 1.82 g cm$^{-3}$), the gravimetric-specific strength is as high as 524.32 N m g$^{-1}$, towering over most common metallic alloy, such as aluminum alloy and titanium alloy (Fig. 6g)[42]. The drastic improvement of strength comes from the diminishment of

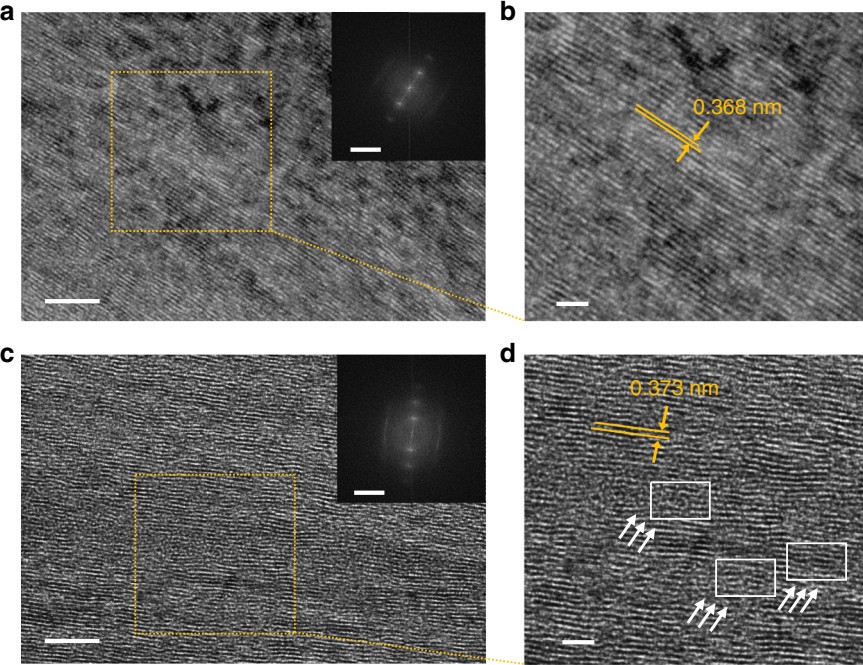

**Fig. 5 Crystalline order of 8% stretching ratio and direct-cast graphene papers. a–d** High-resolution TEM images of 8%-GPs (**a**, **b**) and direct-cast GPs (**c**, **d**), respectively. The insets of (**a**) and (**c**) correspond to the FFT images. Scale bars, 5 nm (**a**, **c**), 2 nm (**b**, **d**), and 5 nm$^{-1}$ (insets of **a**, **c**).

multiscale disorders, which mainly act as stress concentration sites[33,34]. With the plasticization stretching applied, wrinkled graphene sheets are extended to become flat, thus affording effective load transfer and uniform stress distribution. We evaluated the stress distribution of 8%-GPs and direct-cast GPs under tension by Raman spectroscopy (Fig. 6d). G-band has a downshift to lower frequency when tensile strain is applied[43–47]. As shown in Fig. 6d, at a tensile strain of 1%, G-bands of both 8%-GPs and direct-cast GPs have an obvious downshift. However, 8%-GPs show larger downshift of about 4 cm$^{-1}$ compared with that of direct-cast GPs (less than 1 cm$^{-1}$), indicating a more effective load transfer[47]. Moreover, the narrower G-band frequency distribution (1.8 cm$^{-1}$) of 8%-GP implies its more uniform stress distribution, as compared with the wide distribution (4.6 cm$^{-1}$) of direct-cast GPs.

The integration of both ultrahigh strength and stiffness makes our GP a conspicuous structural material. Previous reports on improving the mechanical properties of GPs follows two directions (Fig. 6f)[5–19]. One method is supplementing extra interlayer interactions (e.g., hydrogen bonding and π–π interactions) by blending second phase to facilitate loading transfer[7,8,11,13–15], giving enhanced strength but decreased stiffness since the introduction of second soft phase, as marked in the gray circle in Fig. 6f. For instance, Wan et al.[8] reported a GP with the strength of 0.94 GPa by sequentially bridging the graphene sheets using polymeric chains, whereas its stiffness was only 15.6 GPa, far lower than that of our crystalline GPs (60.27 GPa). The other method improves the stiffness of GPs mainly through increasing short-range interlayer crosslinking[6,10], whereas it ignores the undiminished stress concentration sites by wrinkles, resulting in an unfulfilled strength below 300 MPa. Distinct from these two trends, our IMP stretching method to enhance the crystalline order of GPs simultaneously achieves both ultrahigh strength and stiffness. As compared with artificial nacre materials of inorganic sheets with interlayer polymer binders, GPs also possess a favorable mechanical strength, which is 175% higher than that of polyvinyl acetate/clay composite (400 MPa; Supplementary Fig. 12). Although the wrinkle engineering in our

work cannot be directly used for these 2D rigid sheets, the solvent-intercalation-induced plasticity and the corresponding plastic stretching are possibly useful to improve the structural order and mechanical strength of these laminate composites.

The neat constitution of our GPs endows a spontaneous enhancement in electrical and thermal conductivities. As a comparison, the introduction of second phase increases the electron and phonon scattering centers, thus degrading the electrical and thermal conductivities[49]. Figure 6e shows that the electrical and thermal conductivities are improved gradually with the increased SR. The electrical conductivity of 8%-GPs was measured as $1.09 \times 10^5$ S m$^{-1}$, 62% higher than that of direct-cast GPs ($6.75 \times 10^4$ S m$^{-1}$). Using the steady-state electro-thermal technique in vacuum chamber[48], the thermal conductivity of 8%-GPs was measured as 109.11 W m$^{-1}$ K$^{-1}$ (Supplementary Fig. 10), 158% higher than that of direct-cast GPs (42.34 W m$^{-1}$ K$^{-1}$). Notably, the gravimetric-specific thermal conductivity of 8%-GPs reaches 0.60 W cm$^2$ K$^{-1}$ g$^{-1}$, 122% higher than that (0.27 W cm$^2$ K$^{-1}$ g$^{-1}$) of direct-cast GPs and even 33% higher than that (0.45 W cm$^2$ K$^{-1}$ g$^{-1}$) of the benchmark copper[50].

**Graphene paper laminated composites.** Our continuous IMP stretching strategy is an efficient scalable approach to fabricate strong and multifunctional GPs. To extend the realistic application of GPs, we fabricated an industrial laminated composite with the integration of structural strength and functions, using GP as an independent component as compared with previous solution-processed layer-by-layer assembly[51–53]. The GP/epoxy resin (EP) composites were prepared by the industrial thermal compression technique (Fig. 7a–c). The composite has a laminated structure with alternative GP and EP layers, as shown in SEM images (Fig. 7d–f). Without any surface treatment as carbon fiber composites[54], GP has a good adhesion to EP matrix and exhibits negligible destructive interface like voids or incomplete filling (Fig. 7f), which benefits from the residue oxygen functional groups on GPs. The tensile strength and Young's modulus of the composite consisting of EP and 8%-GPs reach 634 MPa and 25

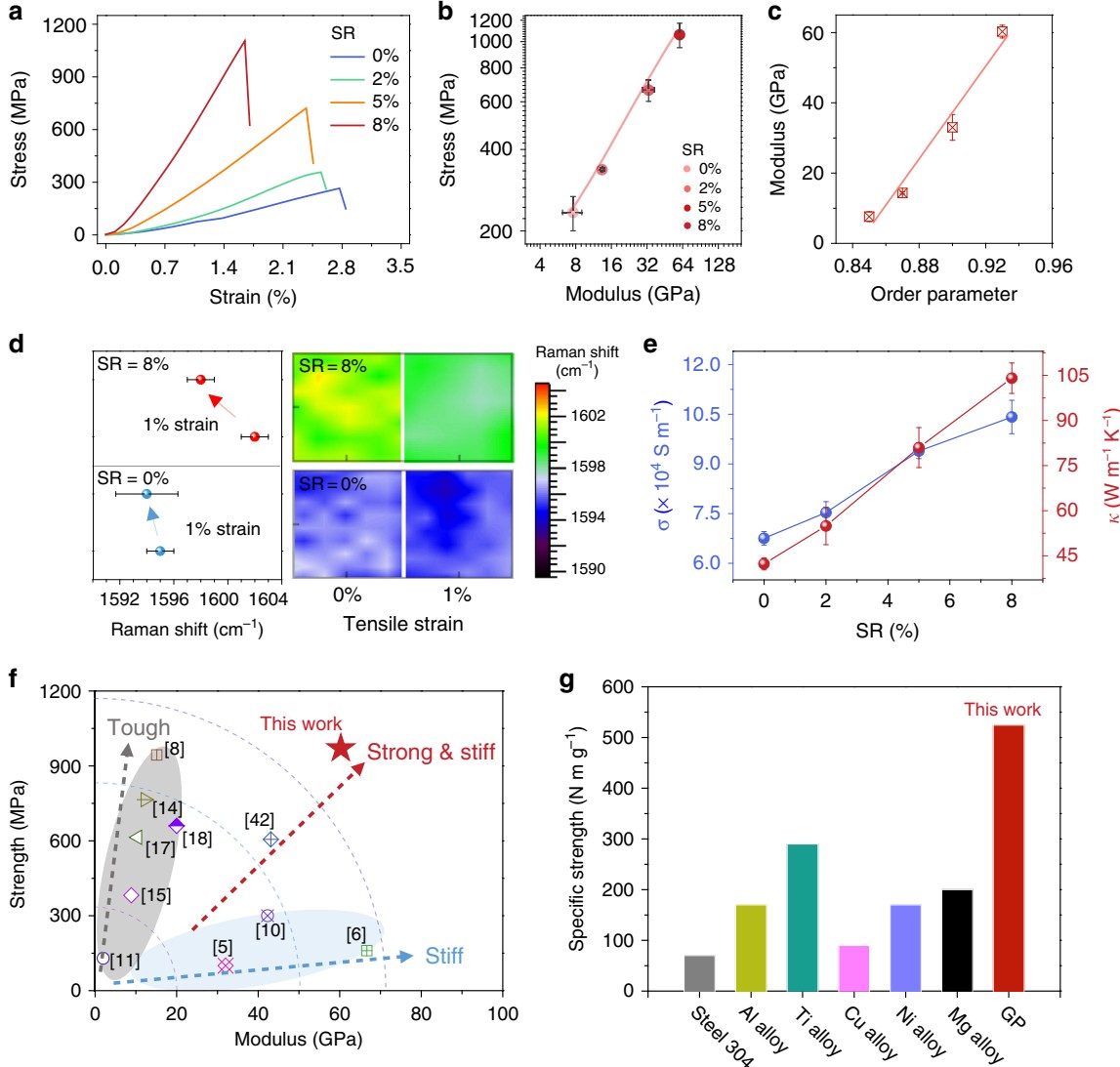

**Fig. 6 Mechanical, electrical and thermal properties of graphene papers. a**, **b** Mechanical strength and modulus of GPs at different *SR*. **c** Relationship between Young's modulus and the order parameter, *f*. **d** Raman shifts of G-band of direct-cast (blue points) and 8%-GPs (red points) at 1% strain. All Raman images collected within an area of 10 × 10 μm². **e** Electrical (*σ*; blue points) and thermal (*κ*; red points) conductivities of GPs at different *SR*. **f** Comparison of strength and modulus of 8%-GPs with previous reported GPs. Data represent means + s.d. **g** Comparison of specific strength of 8%-GPs with conventional metal materials.

GPa, respectively, enhancing the pure EP by 476% in strength and 733% in modulus. We provide an alternative protocol to utilize 2D sheets in advanced composites, taking the assembled papers of 2D sheets with qualified properties as independent components for composite bulks. This method shows a great merit in the large-scale fabrication of composite bulks, which differs from the solution-processed layer-by-layer assembly to prepare composites taking individual 2D sheets as a component[51–53]. Following this protocol, GP/EP composites still hold a superior mechanical strength, 58.5% higher than that of rigid clay-based laminated composite (400 MPa, Supplementary Fig. 12)[48,55–58].

Beyond the mechanical performance, the EP composite exhibits a high electrical conductivity up to $3.1 \times 10^4$ S m$^{-1}$. As a comparison, the laminated composite consisting of direct-cast GPs only has a tensile strength of 190 MPa, Young's modulus of 6 GPa, and electrical conductivity of $1.9 \times 10^4$ S m$^{-1}$. We investigated the electromagnetic interference (EMI) shielding performance and the EMI shielding effectiveness (SE) of the GP/EP composite is about 30−40 dB in 2−18 GHz. The integration

of high mechanical strength, stiffness, toughness, electrical conductivity, and low density (1.4 g cm$^{-3}$) of GP/EP composites has a realistic potential as advanced composites in airplane and aerospace industry.

## Discussion

In summary, we observed a transition from brittleness to plasticity for GO papers by the intercalation of solvent plasticizers. Such a facile plastic state plays a vital role for secondary processing of graphene-based assemblies as the cases of metals and plastics processing. The plastic transition allows the elimination of random wrinkles and the industrial stretching crystallization. The extended conformation of graphene sheets in GPs promotes the orientation and stacking order of GPs, affording a highly crystalline ordered structure. The formation of crystalline structure reduces the stress concentration sites and scattering centers of electrons/phonons, making GPs approach gigapascal strength, high stiffness and excellent conductivities. Our IMP stretching method offers an important supplementary to the previous

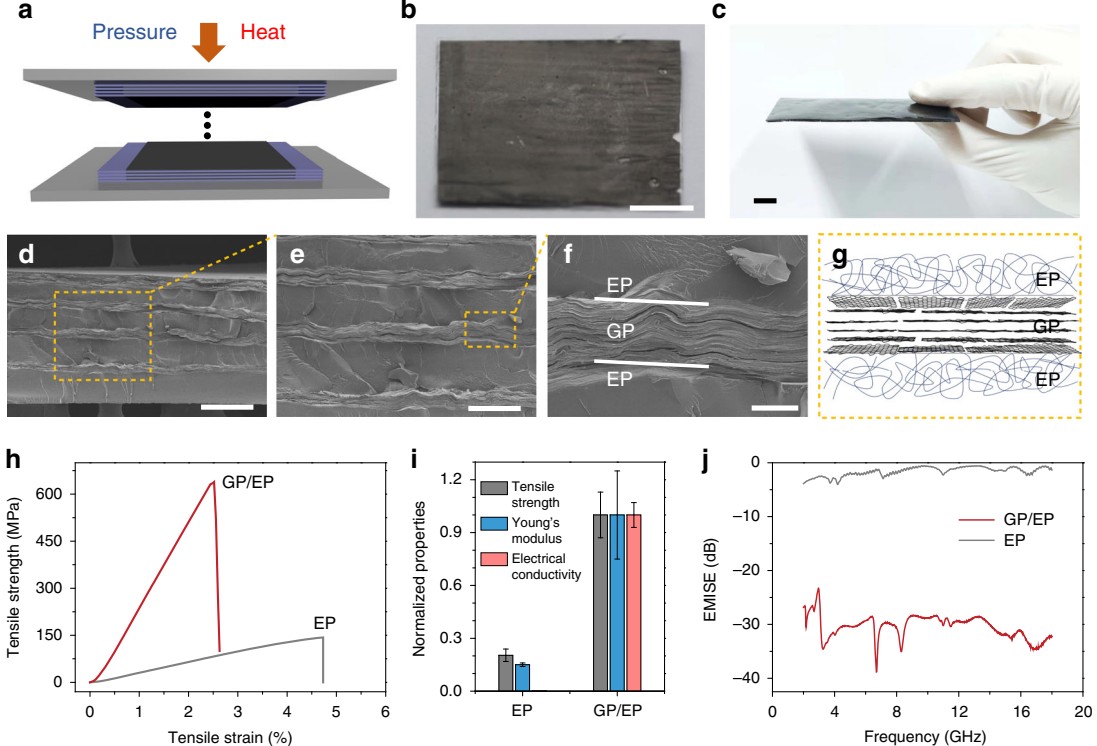

**Fig. 7 Graphene paper laminated composites. a–c** Schematic (**a**) and photographs (**b**, **c**) of the laminated composites consisting of EP and GPs. **d–g** SEM images and schematic of the laminated composite, showing a alternatively laminated structure. **h** Tensile curves of the laminated composites (8%-GP/EP) and pure EP. **i** Comparison of tensile strength, Young's modulus and electrical conductivity between the 8%-GP/EP composite and pure EP. All data are normalized by the properties of 8%-GP/EP composite, which shows average tensile strength, modulus, and electrical conductivity of 540 MPa, 20 GPa, and $2.9 \times 10^4$ S m$^{-1}$, respectively. Error bars are defined as s.d. **j** EMI shielding properties of the 8%-GP/EP composite and pure EP in 2–18 GHz. Scale bars, 10 mm (**b**, **c**), 200 µm (**d**), 100 µm (**e**), 20 µm (**f**).

preparation of GPs, and completes the structural control of GPs starting from the dispersed state to the final solid state. For the convenience of the IMP stretching strategy, we developed a continuous approach for strong GPs that can be compatible with industrial production. The strong and conductive GPs were also extended to fabricate a realistic laminated composite with the integration of high structural properties and excellent functional applications. This facile IMP strategy is suitable to macro-assemblies of other 2D sheets, and would be an important engineering process in the industry.

## Methods

**Fabrication of graphene papers**. Aqueous GO dispersion was purchased from Hangzhou Gaoxi Technology Co. Ltd (www.gaoxitech.com). The average size of the GO dispersion was measured as 12.5 µm and the Zeta potential of aqueous dispersions was tested as −39.72 mV. Typically, aqueous GO solution (7 mg g$^{-1}$) was cast-dried as strip shape at ambient temperature and relatively high humidity to make it dehydrate slowly. Peeling off the dried coating from the substrate obtains freestanding direct-cast GO papers. For the IMP stretching, direct-cast GO papers were fed into the two rollers, keeping paper immersed in ethanol for more than 2 min. The speed ratio was adjusted between the two rollers to tune SR. Followed by reduction by the mixture of HI and ethanol (volume ratio of 1:3) in a tension state on graphite roller, GPs with varied SRs were fabricated.

**Fabrication of laminated composites**. GPs, direct-cast or stretched, and EP resin were alternatively placed on a Teflon plate. Then, applying pressure to 10 MPa and keeping at 80 °C for 12 h can obtain the GP/EP laminated composite.

**Characterization**. Mechanical test was taken on Instron 2344 at a loading rate of 5 mm min$^{-1}$, with strip-like samples of 10 mm × 1 mm (at least three samples). SEM inspections were taken on Hitachi S4800 field emission system. Raman spectra were taken on a Renishaw in Via-Reflex Raman microscopy at an excitation wavelength of 532 nm. XRD profiles were collected on a X'Pert Pro (PANalytical) diffractometer using monochromatic Cu 17 K$_{\alpha1}$ radiation ($\lambda = 1.5406$ Å) at 40 kV.

POM observations were performed on a Nikon E600POL. Zeta potential was taken on a ZET-3000HS. Synchronous SAXS and WAXS tests were carried out in BL16B1 beam line station of Shanghai Synchrotron Radiation Facility (SSRF). Samples for SAXS, 2D WAXS measurements were prepared by aligning papers on the side of quartz plate for the transmission mode. Thermal conductivity was measured (three samples) by the steady-state electro-thermal technique in vacuum chamber, in which temperature profiles were collected by an FLIR T630sc infrared camera with a close-up lens. Electrical conductivity was measured (three samples) using a four-probe method on an electrical transport properties measurement system comprising a Keithley 2400 multiple-function source-meter. The electromagnetic interference performance was tested by a vector network analyzer (ZNB-40, Rohde & Schwarz, Germany).

**Calculation of order parameters**. Quantification of SAXS and WAXS patterns was performed with scattering vector $q$ and azimuthal angle $\varphi$ as coordinates. SAXS was used to characterize the orientation order on tens of nanometer scale. WAXS was applied to evaluate the order parameters at the atomic scale. The orientation of graphene sheets in papers was quantified by the (002) reflection in WAXS patterns.

The orientation was quantified by converting the orientation distribution to a Herman's order parameter, $f$, defined as

$$f = \left\langle \frac{3}{2}\cos^2\varphi - \frac{1}{2} \right\rangle, \tag{1}$$

where $\varphi$ is the azimuthal distribution, and the order parameter can be expanded as

$$f = \int_0^\pi I(\varphi)\left(\frac{3}{2}\cos^2\varphi - \frac{1}{2}\right)\sin(\varphi)d(\varphi). \tag{2}$$

The intensity is normalized according to

$$\int_0^\pi I(\varphi)\sin(\varphi)d(\varphi) = 1. \tag{3}$$

**Numerical simulation**. Based on a representative volume element (RVE) of previously reported DTS chain model, when the tensile force is applied to the RVE, two distinct failure modes may occur, i.e. the GO fracture (mode G) or interlayer sliding (mode I), which is determined by the mechanical properties of the graphene oxide and interlayer links (van der Waals attraction, π−π interaction and hydrogen

bonding), as well as size of graphene oxide and interlayer distance, written as

$$\sigma_s = \min \begin{cases} \frac{\sigma_{cr} h}{2 h_0} & \text{(mode G)} \\ \frac{D s \gamma_{cr}}{2(1+c) l_0} & \text{(mode I)}, \end{cases} \quad (4)$$

where $\sigma_{cr}$ and $\gamma_{cr}$ are the critical tensile stress and critical interlayer shear strain, respectively. $h$ is the thickness of GO, $h_0$ is the interlayer distance, $s = \sinh(l/l_0)$, $c = \cosh(l/l_0)$, and $l$ is the size of RVE. $l_0 = (Dh_0/4G)^{1/2}$ is the length scale of interlayer load transfer through parameters including the interlayer distance, $h_0$, the effective shear modulus $G$, and tensile stiffness of the GO, $D = Yh$, where $Y$ is its tensile modulus.

In order to describe the mechanical behaviors of GOPs with the intercalation of ethanol, we introduce the concepts of self-healable interlayer links-regulated toughening mechanism for GOPs. For the interlayer sliding failure mode, the self-healable interlayer links can repeatedly fail and reconstruct to dissipate a large amount of energy to increase toughness of GOPs. A nonlinear spring-bead model is proposed (Supplementary Fig. 3a). Here, the two GO sheets are modeled as two chains of beads. The intralayer covalent bonds are simplified as linear springs, while the self-healable interlayer links are simplified as the reconstructable springs, and the mass of GO is constricted at the beads. The parameters of the nonlinear spring-bead model can be derived from the experiments. For examples, the intralayer spring constant is derived as $k_1 = Eh/r_1$, and the interlayer spring constant is derived as $k_2 = Gr_1/h_0$, where $E$, $G$, $h$, $h_0$ and $r_1$ are in-plane elastic modulus, interlayer shear modulus, thickness of GO, interlayer distance and original length of the intralayer spring, respectively. In order to describe the wrinkle effect on the mechanical behaviors of GOPs, a nonlinear spring with stiffness constant $k_3$ is introduced (Supplementary Fig. 3a). At first the intralayer spring (spring 1) at the wrinkle region is switched off and the horizontal distance of neighbor beads is $r_2$. Then, after the flattening of wrinkles (i.e. the horizontal distance of neighbor beads becomes $r_1$), the wrinkle spring (spring 3) is failed and intralayer spring (spring 1) is switched on. During tension, a constant speed $v$ is applied to the first bead in Sheet 1 and the last bead in Sheet 2, and the external force acting on the beads is calculated as $F = k_1(u_{12} - u_{11}) + k_2(u_{21} - u_{11})$, where $u_{11}$, $u_{12}$, $u_{21}$ are the displacment of the first bead in sheet 1, second bead in sheet 1 and first bead in sheet 2, respectively. The effective tensile stress (defined as $F/h_0$) and strain (defined as $(u_{n1} - u_{21})/l$, where $l$ is size of RVE) of the RVE are recorded to draw the simulated tensile curve.

The simulated tensile behavior of GOPs with ethanol intercalated in 1.58 nm interlayer spacing. The simulated tensile curve exhibits a typical plastic deformation trend with a high breakage elongation of 10%, which coincides well with the experimental result (Supplementary Fig. 3b). The tensile curve can be divided into three stages. At small strain region (<1%), the stress−strain relation is almost linear corresponding to elastic deformation of the GOPs. Then, at moderate strain region (1−5%), the slope of the stress−strain curve decreases corresponding to the softening effect caused by the wrinkle flattening. Thereafter, at large strain region, the interlayer sliding and reconstruction occurs and the slope of the stress−strain curve increases corresponding to the hardening effect caused by the interlayer links reconstruction. Because of continuous energy dissipation by the interlayer links failure and reconstruction, the plastic limit strain of GOPs is significantly improved.

## Data availability
The data that support the findings of this study are available from the corresponding authors upon reasonable request.

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

## Acknowledgements

We thank the members of staff at Shanghai Synchrotron Radiation Facility (SSRF) for assistance in SAXS and WAXS characterizations. This work is financially supported by the National Natural Science Foundation of China (Nos. 51973191, 51703194, 51533008, 51803177 and 11890674), National Key R&D Program of China (No. 2016YFA0200200), Hundred Talents Program of Zhejiang University (188020*194231701/113), Key Research and Development Plan of Zhejiang Province (2018C01049), the Fundamental Research Funds for the Central Universities (No. 2017QNA4036) and Foundation of National Key Laboratory on Electromagnetic Environment Effects (No. 614220504030717).

## Author contributions

Z.X., Yilun L., and C.G. conceived the research. P.L. and Z.X. designed experiments, analyzed the data, and wrote the manuscript. P.L. and M.Y. did the tensile tests of plasticized films and the EMI tests. P.L. and Yingjun L. did the thermal test. Yilun L., H.Q., and J. Liu did the numerical simulation of the IMP. P.L. and F.M. fabricated freestanding strip-like GO films. P.L., J. Lin, and F.W. analyzed the Raman data.

## Competing interests

The authors declare no competing interests.

## Additional information

**Peer review information** *Nature Communications* thanks the anonymous reviewer for their contributions. Peer review reports are available.

