## [Peer Review File · Nature Communications]

Reviewers' comments:

Reviewer #1 (Remarks to the Author):

Peng Li et al describe a method for manufacturing composites from graphene with high 'vertical' crystalline order of graphene nano assemblies using a continuous assembly system. They address the problem of some composites that have high in-plane strength yet soft in out-of-plane direction. These properties are indeed difficult to combine if the graphene sheets have wrinkles.

The part to the high stiffness in both directions is to have high degree of organization from layer to layer. The authors achieve it using intercalation modulated plasticization (IMP) phenomenon and integrate this method in the roll-to-roll process achieving tensile strength of 1.1 GPa and stiffness of 62.8 GPa) and electrical conductivity of $1.1 \times 10^5 \text{ S m}^{-1}$.

Overall, this reviewer believes that the manuscript has excellent manufacturing aspect. Mechanical properties are also adequate. However, there are substantial omissions and the opportunity for broader impact. The manuscript may be potentially published in Nature Comm provided the following comments are addressed.

1. It seems to me that the improvement of organization is only one part of the answer here. The intercalation modulated plasticization (IMP) technique invented by the authors coupled with the stretching inherent to their method of making graphene paper increases the thermodynamic minimum of binding the polymer to the inorganic sheets. In part they can see it on polarized SAXS measurement. However, the important thing would be to look at FTIR spectrum of the composite under different treatments. I am positive that the authors will see strong change in the bonding between the phases. NMR spectrum can help as well.

2. The mechanical properties obtained here are impressive and deserve due attention. However, I also think that the authors can make a step further and compare their results with other layered composites, such as clay nanosheets. The problem of wrinkles is less important for clays because of their higher rigidity and thus the properties can also rise. One example of that are the layered composites reported from nanosheets of montmorillonites:
<https://science.sciencemag.org/content/318/5847/80/tab-pdf>
that are at least comparable to the mechanical data reported here. Albeit made in small batches. In this aspect the roll-to-roll IMP has clear advantages and can be extended to other systems.

3. Improvement of the crystallinity was also investigated in previous studies on layer-by-layer assembly of graphene and graphene oxide that are directly relevant to the concept of this study:
<https://pubs.acs.org/doi/full/10.1021/acs.chemmater.6b02688>;
<https://pubs.acs.org/doi/pdf/10.1021/nn400972t>;
<https://www.sciencedirect.com/science/article/pii/S0376738814004876>;
<https://pubs.acs.org/doi/abs/10.1021/j100035a005>

Although some of these studies deal with the composites of graphene that the authors are dealing with and with the same question how to organize them from the perspective of crystallinity and 'vertical' organization to attain high mechanical and electrical properties. The difference is that pure graphene paper without the polymer will have high stiffness but the toughness will be low. Many applications including the EMI shielding will require all four parameters: strength, stiffness, toughness, and conductivity. I think the method of Peng Li et al could be applicable to composites and optimization of toughness.

4. For reproducibility the average size of graphene platelets as well as their charge in the used dispersions need to be added to the characterization of the materials.

Reviewer #2 (Remarks to the Author):

The authors present a continuous crystalline graphene papers by a simple intercalation modulated plasticization. The random wrinkles were eliminated. The excellent mechanical and conductive properties are demonstrated. The manuscript was well organized and characterized. However, some reports demonstrated the similar performance or even better, and applicable product can be available. It is necessary to compare it with other common methods. The working principles are also not stated very clearly. Therefore, I don't think it meets the high requirements of Nature Communication, and suggest transferring to scientific report or other specific journals.

Reviewer #3 (Remarks to the Author):

In this manuscript, the authors described a facile intercalation modulated plasticization (IMP) stretching strategy to tune the random wrinkles of graphene sheets into an ordered structure, and then obtain crystalline graphene papers (GPs) that combines gigapascal strength, high stiffness and outstanding thermal/electrical conductivities. The work provides a convenient a continuously fabricate crystalline GPs with a meter-scale length, which can be compatible with industrial production. The result is an enlightening and practical guide for the design of laminated composites with the integration of high structural properties and multifunctionalities. The discussion of the results needs to be improved, therefore, it can be considered for publication but only after major revisions.

1. On Page 5, line 1-2, the paper showed that "But for direct-cast GOPs, the fracture 1 width is only 12 μm with a slippage band width 2 of 2.0 μm (Fig. 2d, e; Supplementary Fig. 8a-d)." However, Fig. 2 gave the XRD profiles of the direct-cast GOP and the plasticized GOP, while Fig. S8 showed the fracture of direct-cast GPs (a-d) and 8%-GPs. It is clear that Fig. 2d and Fig. 8a-d present the properties of GOP and GP. Please check them carefully.

2. On page 5, line 28, the authors described that "WAXS measurements on the section parallel to the stretching direction". But the Fig. 4c displayed completely the opposite direction. Therefore, the authors should indicate the exact directions of the Fig. 4c and S5c, and modify the text accordingly.

3. On page 6, line 13, the authors claimed that "after 2% SR, the Poisson ratio became positive and the wrinkles along the stretching direction became compressed". However, the Poisson ratio couldn't be found the paper. How do the authors give the result? Furthermore, as shown in S5, the wrinkles along the stretching direction should be changed to the wrinkles perpendicular to the stretching direction.

4. On page 6, line 19, What does 8%-GPs stand for? Full name of abbreviation should be explained for the first time.

5. On page 6, line 23-24, and page 7, line 4-5, the description sections lack the corresponding figure tags. Please add them in the revised manuscript.

6. As can be seen in From Fig.2 and Fig. S8, the fractures of a direct-cast and plasticized GOPs exhibited obvious difference, but those of GPs were basically the same. Why? Why did the GPs have the plastic behavior?

7. The discussion in the text didn't match the relative figures, including Fig. 2c, d, Figure 7a-f (page 5, line 16; page 8, line 17). Please check and revise them carefully.

8. On page 8, line 27-28, the authors also mentioned that the GP/EP composite had low density. However, the density of sample wasn't presented. Please add them in the revised manuscript.

9. As shown in Fig. 6 d-e, what do the red and blue points respectively stand for? Please add them in the figures.

10. There are a few typos and grammar errors in the manuscript, and the authors should check carefully and correct them. For instance, "increment" (page 8, line 2)

11. Full name and the abbreviation need to be consistent throughout, such as, "Figure" and "Fig."

Point to point responses to the reviewer comments:

Reviewer # 1 (Remarks to the Author):

Comment 1. Peng Li et al describe a method for manufacturing composites from graphene with high ‘vertical’ crystalline order of graphene nano assemblies using a continuous assembly system. They address the problem of some composites that have high in-plane strength yet soft in out-of-plane direction. These properties are indeed difficult to combine if the graphene sheets have wrinkles.

The part to the high stiffness in both directions is to have high degree of organization from layer to layer. The authors achieve it using intercalation modulated plasticization (IMP) phenomenon and integrate this method in the roll-to-roll process achieving tensile strength of 1.1 GPa and stiffness of 62.8 GPa and electrical conductivity of $1.1 \times 10^5 \text{ S m}^{-1}$.

Overall, this reviewers believes that the manuscript has excellent manufacturing aspect. Mechanical properties are also adequate. However, there are substantial omissions and the opportunity for broader impact. The manuscript may be potentially published in Nature Comm. provided the following comments are addressed.

Response 1: Thanks for your positive comments on the significance of our paper. We have revised the paper as you suggested.

Comment 2. It seems to me that the improvement of organization in only one part of the answer here. The intercalation modulated plasticization (IMP) technique invented by the authors coupled with the stretching inherent to their method of making graphene paper increases the thermodynamic minimum of binding the polymer to the inorganic sheets. In part they can see it on polarized SAXS measurement. However, the important thing would be to look at FTIR spectrum of the composite under different treatment. I am positive that the authors will see strong change in the bonding between the phases. NMR spectrum can help as well.

Response 2: Thanks for your kind suggestion.

We think there is a possible misunderstanding on our paper. The GO and RGO papers prepared in this paper are neat GO and RGO sheets, without any presence of polymer

binders. This pure constitution of our paper is different from the nacre-like papers with polymer binders and inorganic bricks. In these nacre composites, FTIR has been used to confirm the formation of coordination/covalent links between metal ions of inorganic bricks and oxygen functional groups of polymers (Science, 2007, 318(5847): 80-83). But in our system, the plastic-stretching process did not bring any change of the composition of GO and RGO papers. It only gave a rearrangement of GO and RGO sheets to have higher crystallinity, that is the very focus of our paper. To prove our theory, SAXS is not the only characterization and we also used WAXS, section TEM, optical microscopy and SEM tracking, and density measurement to systematically evaluate the multiscale structure, that matches the responding mechanical properties.

Comment 3. The mechanical properties obtained here are impressive and deserve due attention. However, I also think that the authors can make a step further and compare their results with other layered composites, such as clay nanosheets. The problem of wrinkles is less important for clays because of their higher rigidity and thus the properties can also rise. One example of that are the layered composites reported from nanosheets of montmorillonites: <https://science.sciencemag.org/content/318/5847/80/tab-pdf>. That are at least comparable to the mechanical data reported here. Albeit made in small batches. In this aspect the roll-to-roll IMP has clear advantages and can be extended to other systems.

Response 3: We supplemented a comparison of our neat graphene paper and its laminated composites with previous results of other layered composites, as shown in Supplementary Fig. 12 in the revised manuscript. Corresponding discussion was also added in the main text.

Comparison of neat graphene paper with previous layered composites (page 8, line 12-18):

As compared with artificial nacre materials of inorganic sheets with interlayer polymer binders, GPs also possess a favorable mechanical strength, which is 175% higher than that of PVA/clay composite (400 MPa; Supplementary Fig. 12). Although

the wrinkle engineering in our work cannot directly used for these 2D rigid sheets, the solvent-intercalation induced plasticity and the corresponding plastic stretching are possibly useful to improve the structural order and mechanical strength of these laminate composites.

Comparison of GP/EP composites with previous layered composites (page 9, line 10-16):

We provide an alternative protocol to utilize 2D sheets in advanced composites, taking the assembled papers of 2D sheets with qualified properties as independent components for composite bulks. This method shows a great merit in the large scale fabrication of composite bulks, which differs from the solution-processed layer-by-layer assembly to prepare composites taking individual 2D sheets as a component⁵¹⁻⁵³. Following this protocol, GP/EP composites still hold a superior mechanical strength, 58.5% higher than that of rigid clay-based laminated composite (400 MPa, Supplementary Fig. 12)^{48, 55-58}.

Supplementary Figure 12. Comparison of tensile strength and Young's modulus for two-dimensional nanosheets-based composites, which were prepared by casting/coating from dispersions (labeled as D) and layer-by-layer assembly (labeled by L).

Comment 4. Improvement of the crystallinity was also investigated in previous studies on layer-by-layer assembly of graphene and graphene oxide that are directly relevant to the concept of this study:

<http://pubs.acs.org/doi/full/10.1021/acs.chemmater.6b02688>;

<http://pubs.acs.org/doi/pdf/10.1021/nn400972t>;

<http://www.sciencedirect.com/science/article/pii/S0376738814004876>;

<http://pubs.acs.org/doi/abs/10.1021/j100035a005>

Although some of these studies deal with the composites of graphene that the authors are dealing with and with the same question how to organize them from the perspective of crystallinity and ‘vertical’ organization to attain high mechanical and electrical properties. The difference is that pure graphene paper without the polymer will have high stiffness but the toughness will be low. Many applications including the EMI shielding will require all four parameters: strength, stiffness, toughness, and conductivity. I think the method of Peng Li et al could be applicable to composites and optimization of toughness.

Response 4: Thanks for your comment. We have added these closely related references and discussions in the revised manuscript (please see the highlights on page 9, line 1-3; page 9, line 10-16; page 9, line 21-24; ref. 48, 51-53, 55-58). As you insightfully predict, we are extending this method to the graphene based composite papers and trying to develop graphene composite paper materials with qualified performances and multifunctionalities.

Comment 5. For reproducibility the average size of graphene platelets as well as their charge in the used dispersions need to be added to the characterization of the materials.

Response 5: Thanks for your advice. We have added the size information and their Zeta potential of the raw GO materials in the revised manuscript (please see the highlights on page 10, line 14-15; Supplementary Fig.1).

Supplementary Figure 1. SEM image (a) and corresponding size distribution (b) of GO. Scale bar, 50 μm (a)

Reviewer # 2 (Remarks to the Author):

Comment. The authors present a continuous crystalline graphene papers by a simple intercalation modulated plasticization. The random wrinkles were eliminated. The excellent mechanical and conductive properties are demonstrated. The manuscript was well organized and characterized. However, some reports demonstrated the similar performance or even better, and applicable product can be available. It is necessary to compare it with other common methods. The working principles are also not stated very clearly. Therefore, I don't think it meets the high requirements of Nature Communication, and suggest transferring to scientific report or other specific journals.

Response: Thanks for your insightful comments.

In this work, we have achieved three important advances:

(1) Graphene papers with both ultrahigh mechanical strength (1.1 GPa) and high stiffness (62.8 GPa) have been continuously prepared, together with many intriguing functionalities. This work directs another important trend of strong and stiff graphene papers for potential structural uses, which is distinct from the recent trend of strong and tough graphene papers. We supplemented the comparison with recently advances (Fig. 6f) and confirmed the superiority of our graphene papers. We also gave a model verification to fabricate strong, stiff and multifunctional laminated composites using our graphene papers.

Fig. 6 f Comparison of strength and modulus of 8%-GPs with previous reported GPs.

(2) A facile yet effective method, that is intercalation modulated plasticization, has been initiated to realize the scalable production of ultrastrong graphene papers. Importantly, this method can proceed in a continuous manner and match the industrial production line of plastic films. To clarify your concern, we give a specific comparison between previous methods and ours (Supplementary Tab. 1), in correlation with the overall performances of prepared materials (Fig. 6f). In short, our method does not compete with other preparation methods, but an important supplementary to them, which completes the structural control of graphene papers starting from the dispersed state to the final solid state.

The comparison of the previous methods to prepare graphene papers see page 4, line 4-8; page 10, line 2-4:

Different from other previous methods to improve the alignment of GOPs and GPs, such as infiltration-aided assembly and centrifugation sparging^{5, 18}, our method gives an alternative post-strategy to engineer the wrinkles spontaneously formed in the final drying, which has been ignored previously, and completes the structural control of graphene papers starting from the dispersed state to the final solid state.

Supplementary Table 1. Comparison of the GPs synthesized by different methods. Our continuous IMP stretching method supplements the structural control of GPs in a solid plastic state, which completes the whole process of GPs from liquid dispersions to the final solid.

Reference	Strength (MPa)	Young's modulus (GPa)	Preparation	Additive interlayer interaction
8%-GP	1100	60.27	Continuous IMP stretching; Solid processing	No
Direct-cast GP	234.16	7.6	Cast; Solution state	No
5	100	32	Vacuum-assisted filtration; Solution state	No
6	160	66.67	Vacuum-assisted filtration; Solution state	Borate crosslinking
7	178.96	84.84	Vacuum-assisted filtration; Solution state	PEI crosslinking
8	944	15.134	Vacuum-assisted filtration; Solution state	π - π and covalent crosslinking
10	300	42.3	Vacuum-assisted filtration; Solution state	No
11	129.6	1.96	Vacuum-assisted filtration; Solution state	π conjugated crosslinking
14	765	12.299	Cast; Solution state	CNC; Topological design
15	382	8.863	Cast; Solution state	No
17	614	10.406	Cast; Solution state	No
18	660	20	Centrifugal spraying; Solution state	No
40	606	43.1	Vacuum-assisted filtration; Solution state	No

(3) We proposed the new intercalation modulated plasticization mechanism to eliminate the random wrinkles of graphene and achieve the highly crystallinity of their papers. The finding of intercalation modulated plasticization reveals the existence of plastic state of graphene oxide, which is an extremely useful processing state for any materials. The intercalation modulated plasticization facilitates the forthcoming controllable production of graphene materials, extending from graphene papers (in this paper) to fibers, aerogels and other delicate nanostructures (got proven in our other works). We intuitively think that the plastic state in this work can be generalized to other 2D sheets and other 1D nanoparticles.

As for the mechanism, we also theoretically calculated the van der Waals interaction between adjacent GO layers as intercalation occurs (Supplementary Fig. 2). With ethanol intercalated into the GOPs, the van der Waals attraction degrades, thus endowing the interlayer sliding and plastic deformation. We rationalized the intercalation modulated plasticization by a nonlinear spring-bead model with the concepts of self-healable interlayer interaction based on a previously reported deformable tension-shear chain model (Supplementary Fig. 3; *J. Mech. Phys. Solids* 2012, 60(4): 591-605; *J. Mech. Phys. Solids* 2014, 70: 30-41). As the interlayer spacing expands after solvent intercalation, the failure and reconstruction of the weakening interlayer links continuously occur in balance, contributing to the plastic strain of GOP as the dislocation gliding in metals. Based on the model, we took a

numeric simulation of the tensile behavior of GOPs with ethanol intercalated. The numeric simulated tensile curve coincides well with the experimental result as shown in Supplementary Fig. 3b.

The mechanism analysis and simulations of the intercalation modulated plasticization see page 4, line 27-30; page 5, line 1-6:

The supplemented results: According to Lifshiz's formula, the expanding interlayer weakens the van der Waals interaction by intercalating plasticizers and activates the sheet sliding under tension (Fig. 2c; Supplementary Fig. 2)²⁵, in analogy with the thermal plastic mechanism of linear polymers²⁶. We rationalized the elastic-plastic transition by a nonlinear spring-bead model with the concept of self-healable interlayer interactions (for example, hydrogen bonding), based on a previously reported deformable tension-shear chain model (Supplementary Fig. 3)^{27,28}. From this model, their failure and reconstruction of interlayer links initiating at the end and then propagating to the middle of laminated GO sheets afford the plastic deformation. As the interlayer spacing expands after solvent intercalation, the failure and reconstruction of the weakening interlayer links continuously occur in balance, contributing to the plastic strain of GOP as the dislocation gliding in metals. As a result, a large amount of energy is dissipated and the plastic limit strain is significantly increased as observed in tensile tests of GOPs (Fig. 2b). We numerically simulated the tensile behavior of GOPs with ethanol intercalated in 1.58 nm interlayer spacing. The simulated tensile curve exhibits a typical plastic deformation trend with a high breakage elongation of 10%, which coincides well with the experimental result (Supplementary Fig. 3b).

Supplementary Figure 2. The correlation between van der Waals interaction and interlayer distance. As the interlayer distance increases, the van der Waals attraction degrades gradually. The calculation of van der Waals interaction is based on the Lifshitz's formula as follows:

$$W_{vdw}(d) = -\frac{H}{12\pi} \left(\frac{1}{d^2} + \frac{1}{(d+2t)^2} - \frac{1}{(d+t)^2} - \frac{1}{(d+t)^2} \right)$$

where H is Hamaker constant (3.72×10^{-20} J for ethanol)²⁵, d is the interlayer distance, and t is the thickness of GO sheet.

Supplementary Figure 3. Numeric simulation of the IMP. **a** Non-linear spring-bead model of the plasticization of GOPs. Here, the two GO sheets are modeled as two chains of beads. The intralayer covalent bonds are simplified as linear springs, while the self-healable interlayer interactions are simplified as the reconstructable springs, and the mass of GO is constricted at the beads. **b** The numeric simulated tensile curve of the plasticized GOP with ethanol, showing accordance with the experimental result. **c** Schematic diagram of the process of plastic deformation. The interlayer interaction in dried GOPs is powerful that the interlayer sliding is difficult to occur. As the interlayer spacing expands after solvent intercalation, the interlayer sliding is activated, and the failure and reconstruction of the weakening interlayer crosslinks continuously occur in balance, contributing to the plastic strain of GOP as the dislocation gliding in metals.

In a conclusion, we ponder that these advances can give a fresh sight to consider

the production and applications of graphene materials, nacre-like materials, 2D sheets and other nanoparticles. As a result, we ask for your re-consideration to accept our paper in Nature Communications for its broad readers.

Reviewer # 3 (Remarks to the Author):

Comment 1: In this manuscript, the authors described a facile intercalation modulated plasticization (IMP) starching strategy to tune the random wrinkles of graphene sheets into an ordered structure, and then obtain crystalline graphene papers (GPs) that combines gigapascal strength, high stiffness and outstanding thermal/electrical conductivities. The work provides a convenient a continuously fabricate crystalline GPs with a meter-scale length, which can be compatible with industrial production. The result is an enlightening and practical guide for the design of laminated composites with the integration of high structural properties and multifunctionalities. The discussion of the results needs to be improved, therefore, it can be considered for publication but only after major revisions.

Response 1: Thanks for your positive evaluation. We have revised the manuscript as you suggested.

Comment 2. On page 5, line 1-2, the paper showed that “But for direct-cast GOPs, the fracture width is only 12 μm with a slippage band width 2.0 μm (Fig. 2d, e; Supplementary Fig. 8a-d).” However, Fig. 2 gave the XRD profiles of the direct-cast GOPs and the plasticized GOP, while Supplementary Fig. 8 showed the fracture of direct-cast GPs (a-d) and 8%-GPs. It is clear that Fig. 2d and Fig. 8a-d present the properties of GOP and GP. Please check them carefully.

Response 2: Thanks for your correction. Fig. 2d, e shows the fracture of GOPs. Supplementary Fig. 11 shows the fracture of GPs. We have corrected the error in the revised manuscript (please see the highlights on page 5, line 11-13).

Comment 3. On page 5, line 28, the authors described that “WAXS measurements on the section parallel to the stretching direction”. But the Fig. 4c displayed completely

the opposite direction. Therefore, the authors should indicate the exact directions of the Fig. 4c and S5c, and modify the text accordingly.

Response 3: Thanks for your kind reminder. We have corrected the schematic diagram and added explanation in the corresponding figure label (Fig. 4c).

Comment 4. On page 6, line 13, the authors claimed that “after 2% SR, the Poisson ratio became positive and the wrinkles along the stretching direction became compressed”. However, the Poisson ratio couldn't be found the paper. How do the authors give the result? Furthermore, as shown in S5, the wrinkles along the stretching direction should be changed to the wrinkles perpendicular to the stretching direction.

Response 4: In the stretching process of wrinkled graphene papers, the Poisson ratio has a negative value at first for the flatten effect of wrinkles and turns to the normal positive value for the buckling effect. This phenomenon has been clearly demonstrated in the recent paper (Nat. Commun. 2019, 10, 2446) and we related this intrinsic feature with our observations under optical microscopy. We added related discussion in the revised main text (please see the highlights on page 6, line 19-28).

To clear your confusion on the wrinkle direction, we give a clear definition of the wrinkles type as type I (along the stretching direction) and type II (perpendicular to the stretching direction), as shown in Supplementary Fig. 6a. We also revised the related discussion in the text (please see the highlights on page 5, line 20; page 6, line 6-8).

Supplementary Figure 6. The surface feature of fractured GOPs. **a** Schematic diagram of the two types of wrinkles. For clarify, the wrinkles along the stretching direction are defined as type □ wrinkles, and the ones perpendicular to the stretching direction are defined as type □ wrinkles. **b-e** Surface wrinkled textures of direct-cast GOPs, showing that these wrinkles remain even after elastic breakage. **f-i** Surface features of plasticized GOPs. The gradually emerged aligned texture during IMP stretching suggests that graphene sheets are stretched to extended state. Scale bar, 200 μm (**b, f**), 50 μm (**c, g**), 20 μm (**h**), 5 μm (**d, i**), and 500 nm (**e**)

Comment 5. On page 6, line 19, What does 8%-GPs stand for? Full name of abbreviation should be explained for the first time.

Response 5: Thanks very much for your kind reminder. 8%-GPs means GPs with IMP stretching ratio of 8%. We have checked all the abbreviations in the revised manuscript (please see the highlights on page 4, line 2-4).

Comment 6. On page 6, line 23-24, and page 7, line 4-5, the description sections lack the corresponding figure tags. Please add them in the revised manuscript.

Response 6: We have complemented the corresponding Figure tags.

Comment 7. As can be seen in From Fig. 2 and Fig. S8, the fractures of a direct-cast and plasticized GOPs exhibited obvious difference, but those of GPs were basically the same. Why? Why did the GPs have the plastic behavior?

Response 7: Thanks for your comment.

The plasticity of GOPs is activated by the solvent intercalation (Fig. 2). We have supplemented a theoretic description on the intercalation induced plasticity by theoretic calculations and modeling, as shown in Supplementary Fig. 2, 3. This is the very reason that direct-cast GOP (in dry state) are brittle and plasticized GOPs (wet by solvent) are plastic. But, GPs we tested are dried without any solvent intercalation, therefore, exhibit basically the same fracture attributes.

As shown in Supplementary Fig. 11, the facture difference between direct-cast GP and plasticization stretched 8%-GP reflects the structural rearrangement. Higher crystallinity generates more brittle behavior. The loosely piled graphene sheets in direct-cast GP are easier to slide, causing a larger slippage band width (2 μm) than that (0.73 μm) of 8%-GP with higher crystallinity. This fracture difference is in according with the mechanical tests as shown in Fig. 6a.

Comment 8. The discussion in the text didn't match the relative figures, including Fig. 2c, d, Figure 7a-f (page 5, line 16; page 8, line 17). Please check and revise them carefully.

Response 8: We have checked and made corrections in the revised manuscript.

Comment 9. On page 8, line 27-28, the authors also mentioned that the GP/EP composite had low density. However, the density of sample wasn't presented. Please add them in the revised manuscript.

Response 9: Thanks for your kind reminder. The density of GP/EP composite is tested as 1.4 g/cm^3 and we have added it in the revised manuscript (please see the highlights on page 9, line 22).

Comment 10. As shown in Fig.6 d-e, what do the red and blue points respectively stand for? Please add them in the figures.

Response 10: Thanks for your comment. The blue and red points in Fig. 6d represent the Raman shift of direct-cast (SR=0%) GPs and 8%-GPs with IMP SR=8%, respectively. The blue and red points in Fig. 6e stand for the electrical conductivity and thermal conductivity of GPs with variable IMP stretching ratio. We have modified the figures and added the corresponding explanations.

Comment 11. There are a few typos and grammar errors in the manuscript, and the authors should check carefully and correct them. For instance, “increment” (page 8, line 2)

Response 11: We have checked the typos and grammar errors as careful as we can.

Comment 12. Full name and the abbreviation need to be consistent throughout, such as, “Figure” and “Fig.”

Response 12: We have checked the full names and abbreviations throughout the revised paper.

REVIEWERS' COMMENTS:

Reviewer #1 (Remarks to the Author):

I believe the authors made thoughtful edits in the manuscript that was already well written. I agree with the comments about the polymer phase although I am pretty sure that FTIR would give useful information about molecular scale deformation patterns. I would not be completely positive that the deformation process is just sliding the platelets on top of each other. Anyway, the work makes substantial advance manufacturing and materials science aspects of the layered composites. I support its publication in Nature Communications.

Reviewer #3 (Remarks to the Author):

The revisions have been carried out in a satisfactory manner. So, I recommend the publication of this manuscript in its present form.

Response to reviewer`s comments:

Reviewer # 1 (Remarks to the Author):

Comment. I believe the authors made thoughtful edits in the manuscript that was already well written. I agree with the comments about the polymer phase I am pretty sure that FTIR would give useful information about molecular scale deformation patterns. I would not be completely positive that the deformation process is just sliding the platelets on top of each other. Anyway, the work makes substantial advance manufacturing and materials science aspects of the layered composites. I support its publication in Nature Communications.

Response: Thanks for your comment.

As you predicted, the deformation process in the plasticized GO papers is not the neat sliding between laminated GO sheets. This process also includes the flattening of random wrinkles and the deformation of wrinkle network in a much large scale. We emphasized the sliding mechanism because the intercalation activates the mobility of GO sheets that allows the rearrangement to achieve high crystallinity. Therefore, we added the corresponding discussions in the revised manuscript to fully analyze the deformation process (page 5, line 13-14; page 6, line 8-10).

Reviewer # 3 (Remarks to the Author):

Comment. The revisions have been carried out in a satisfactory manner. So I recommend the publication of this manuscript in its present form.

Response: We appreciate the reviewer`s positive comment.